# In Vivo Study of Local and Systemic Responses to Clinical Use of Mg–1Ca Bioresorbable Orthopedic Implants

**DOI:** 10.3390/diagnostics12081966

**Published:** 2022-08-14

**Authors:** Răzvan Adam, Iulian Antoniac, Silviu Negoiță, Cosmin Moldovan, Elena Rusu, Carmen Orban, Sorin Tudorache, Tudor Hârșovescu

**Affiliations:** 1Department of Orthopedics and Traumatology, Elias Emergency University Hospital, 011461 Bucharest, Romania; 2Department of First Aid and Disaster Medicine, Faculty of Medicine, Titu Maiorescu University of Buchrest, 040051 Bucharest, Romania; 3Faculty of Materials Science and Engineering, University Politehnica of Bucharest, 313 Splaiul Independetei, 060042 Bucharest, Romania; 4Department of Anesthesia and Intensive Care, Clinical University Emergency Hospital Elias, 17 Marasti Blvd., 011461 Bucharest, Romania; 5Department of Anesthesia and Intensive Care, Carol Davila University of Medicine and Pharmacy, 020021 Bucharest, Romania; 6Department of Clinical Sciences, General Surgery, Faculty of Medicine, Titu Maiorescu University of Buchrest, 040051 Bucharest, Romania; 7Department of General Surgery, Witting Clinical Hospital, 010243 Bucharest, Romania; 8Department of Preclinical Sciences, Biochemistry, Faculty of Medicine, Titu Maiorescu University of Buchrest, 040051 Bucharest, Romania; 9Intensive Care Unit Department, Monza Oncology Hospital, 013812 Bucharest, Romania; 10Department of Preclinical Sciences, Anatomy and Embryology, Faculty of Medicine, Titu Maiorescu Univesity of Bucharest, 040051 Bucharest, Romania

**Keywords:** resorbable, Ma-1Ca, alloy, implant, in vivo

## Abstract

(1) Background: Resorbable Mg-based implants represent a new direction in orthopedic surgery but have some drawbacks, such as their rapid biodegradation and increased rate of corrosion. Some in vitro studies hypothesized that tissue necrosis, incision dehiscence, risk of gas embolization in vital organs, interference with coagulation processes, and trophocyte viability impairment can occur. (2) Methods: We conducted an in vivo study on ten rabbit cases, in two groups; group one, consisting of six cases, received cylindrical implants of Mg–1Ca alloy in tibial intramedullary bone tissue. Group two, consisting of four cases, received Mg–1Ca parallelepiped implants, in the thigh muscular tissue. We recorded and compared weight (preoperatively and at 2, 4, and 6 weeks postoperatively), complete blood count, serum electrolytes, liver and kidney functional markers, and coagulation parameters, prior to and at 6 weeks after surgery. Local evolution was assessed radiologically and with tissue biopsies with complete pathology analysis. (3) Results: All biological markers and clinical evolution were favorable, showing good integration of the implants with none of the local or systemic signs of degradation. (4) Conclusions: Our study shows that the clinical use of Mg–1Ca bioresorbable alloys can be safe as none of the cited local or systemic complications have been identified.

## 1. Introduction

The continuous development and modernization of surgical techniques, methods, and materials lays at the very foundation of modern medicine. As such, bioresorbable osteosynthesis materials are regarded as the new frontier in orthopedic surgery. These are mainly used in small bone injuries or ligamentoplasty fixation systems. Resorbable implants are characterized by two main elements: biodegradability, whereby under the surrounding biological environment they will turn into compounds that will be absorbed by the body, and the ability to be biocompatible, thus minimizing the risk of rejection [1]. The orthopedic implant must be mechanically resistant until callus formation, and by resorption must allow the gradual transfer of load-bearing capacity to the bone, avoiding stress shielding and reducing the risk of post-implant fracture. In the case of an orthopedic implant, the resorption process should ideally occur through biodegradation and bone remodeling. It should exhibit osteoconductive properties, represented by the material’s ability to stimulate the growth of bone tissue from the host bone to the implant and the formation of new, adherent bone on the surface of the material (bone adhesion/osseointegration) [2]. Thus, between the surface of the material and the bone tissue, no other tissue should be identified. Resorbable implants made of biopolymers or composite/ceramic materials are now commonly used.

The use of resorbable metallic materials, such as magnesium alloys, represents a new direction of development in orthopedic surgery. Mg-based implants show mechanical properties close to the mechanical values of a healthy bone. They have a density of 1.7–2.0 g/cm^3^, close to natural bone density (1.8–2.1 g/cm^3^) and an elastic modulus of approximately 45 GPa, close to that of natural bone (10–40 GPa). Comparatively, the elastic modulus of usual metallic implants is 100–200 GPa [3].

The main drawbacks of Mg alloys are rapid biodegradation and increased rate of corrosion in the body at a physiological pH of 7.4 to 7.6. Rapid corrosion will lead to loss of mechanical capabilities, the bending strength being able to decrease in a first stage of erosion from 652 MPa to 390 MPa [4,5].

In vitro corrosion studies, also showed that rapid corrosion will increase pH in the corrosion environment, thus producing an alkalizing effect of peri-implant tissues with cytotoxic effect on dendritic line cells. According to ISO 10993-5:2009, a reduction of cell viability of more than 30% is considered a cytotoxic effect [6]. Hydrogen gas is released during the corrosion process of Mg–Ca. Increased hydrogen production rate corresponding to a high corrosion rate can further lead to the formation of gas bubbles in the surrounding tissue. Based on these observations it was hypothesized that tissue necrosis, surgical incision dehiscence, and risk of gas embolization in vital organs can occur [7]. Also, some in vitro studies have hypothesized interference with the clotting process and impairment of trophocyte viability [3].

Magnesium corrosion in solutions follows these reactions: Mg + H_2_O → Mg(OH)_2_ + H_2_ (general reaction); Mg → Mg^2+^ + 2e^−^ (anodic reaction); 2H_2_O + 2e^−^ → H_2_ + 2OH^−^ (cathodic reaction); Mg^2+^ + 2OH^−^ → Mg(OH)_2_ (product formation) [8].

One method of minimizing these disadvantages is to use a magnesium alloying process. To be used in medical applications, alloys must be made of non-toxic elements. Magnesium–calcium alloys are an optimal solution, being composed of two elements, naturally present in the human body, both with positive biological effects [9]. Magnesium has osteoconductive properties, promoting the formation of new bone by stimulating the adhesion of osteoblastic cells, this leading to the idea that formation of hard callous will be induced at the fracture site. Calcium is an important component of bone tissue and is an essential element in biochemical processes [10]. It has a low density of 1.55 g/cm^3^, which enables Mg–Ca alloys to achieve a density similar to bone, which is 1.8–2.1 g/cm^3^. Also, Mg has the potential to improve the incorporation of calcium into bone. Mg–Ca alloys are the most used and researched alloys, the solubility of Ca in Mg is 1.34 wt% [4].

Mg–Ca alloys have a binary microstructure, consisting of a primary phase, α Mg, and a eutectic secondary phase, with lamellar structure, composed of Mg_2_Ca [11]. From the point of view of electrochemical behavior, the Mg_2_Ca phase is more active than the α Mg phase, thus having a cathodic function with respect to magnesium, this difference leading to the formation of a galvanic circuit. The corrosion process of Mg–Ca alloys is the result of this phenomenon, due to the difference of electrochemical potential between the primary phase α Mg and the secondary phase Mg_2_Ca [3].

The amount of calcium in the alloy will influence its microstructure and, implicitly, the mechanical properties and its corrosion resistance. The results presented in the literature, obtained on alloys with low calcium concentrations, such as 1, 2, and 3%, were better than those with high concentrations, of 5–20% Ca, both in terms of mechanical properties and corrosion rate [12,13]. Of these, Mg–Ca 1% alloys were considered to be optimal for orthopedic implants [10].

In vitro studies have demonstrated the osteoconductive and osteoinductive capacity of Mg–1Ca alloys [11]. In some in vitro studies, it has been observed that, during the corrosion process, a hydroxyapatite layer is formed on the surface of the alloy, having a positive role on bone cell adhesion. Further, according to Zreiqat et al. [14], magnesium ions resulting from the degradation of the Mg–1Ca alloy could increase cell adhesion of HBDC (Human Body Derived Cells) in cell cultures. At the level of HBDC cells grown on the surface of the alloy, increased levels of α5b1 and b1 integrin receptors were discovered, which play a role in mediating cell adhesion to the surface of biomaterials. Thus, it is assumed that the increased viability expressed by Mg–1Ca alloys in cell cultures can be explained by facilitating the early response of osteoblasts, and it can be inferred that the release of Mg2^+^ ions from the alloy can stimulate bone cell adhesion and proliferation, being raised the hypothesis of the positive role of Mg–1Ca alloys in bone healing [9,15].

To elucidate these hypotheses from in vitro studies, we decided to conduct an in vivo study to evaluate the clinical, local, and systemic effects of implants made of resorbable Mg–1Ca metal alloy. In vivo studies represent a higher level of research and can be assimilated as precursors to phase 1 clinical trials. In vitro studies, regardless of their quality and complexity, cannot provide concrete data on the clinical response of a biomaterial when implanted in a living organism [12]. The differences between those two environments could be explained by the homeostasis function of living systems in organisms, whereby they compensate for changes induced by external factors. This homeostatic capacity cannot be reproduced in culture media, regardless of their composition.

## 2. Materials and Methods

During this in vivo study, ten cases of European domestic rabbit, *Oryctolagus Cuniculus*, adult, with a minimum weight of 3.5 kg, were used. The study was carried out in accordance with the European legislation, the European Convention for the Protection of Vertebrate Animals used for Experimental and other Scientific Purposes adopted in 1986 in Strasbourg. The European legislation is ratified in Romania by law no. 305/2006, amended and supplemented by law no. 9/2008. At the same time, we have complied with Directive 2010/63/EU of the European Parliament on the protection of animals used for scientific purposes. The study was carried out with the approval of the Ethics Committee for Scientific Research of “Carol Davila” University of Medicine and Pharmacy, Faculty of Medicine, Bucharest, Romania (registered under no 92/20.04.2016, code PO-35-F-03). Animal care and clinical evaluation were performed according to Federation of European Laboratory Animal Science Associations (FELASA) guidelines and recommendations [16].

The rabbit cases were divided in two groups. In Group 1, consisting of six cases, numbered from 1 to 6, we implanted samples of Mg–1Ca alloy into the tibial intramedullary bone tissue. In Group 2, consisting of four cases, numbered from 7 to 10, we implanted samples of Mg–1Ca alloy in the muscular tissue of the thigh.

For group one, cylindrical implants of Mg–1Ca alloy with a density of 1.78 g/cm^3^ and dimensions of 40 mm length and 3.2 mm diameter, were used (Figure 1a). For group two, parallelepiped samples, 1.2 cm/1.2 cm/0.2 cm, were used (Figure 1b). The exposed surface of the sample was 3.84 cm^2^.

For comparison and providing contrast observations we used the rabbits in Group 1 and 2 as control cases, in the following manner: for assessing and comparing characteristics regarding the reactions related to biocompatibility of the Mg–1Ca alloy we used an austenitic steel rod implant positioned in the centro-medullary part of the contralateral limb. On these samples we conducted imagistic and biological tests, including histological determinations, to build a comparison model of the steel implant for assessing the level of osteointegration and osteoconductive capabilities of the Mg–1Ca alloy. For assessing and comparing characteristics such as hydrogen production, biodegradation, stimulation of bone tissue forming and remodeling, and bone integration of the Mg–1Ca alloy, as a contrast we used tissue bone and muscle fragments harvested from the same limb as the Mg–1Ca alloy implant but within a safe distance, to avoid contamination of the samples.

We decided to use austenitic steel as comparison material for its non-oxidative properties as well as for being completely inert to surrounding tissues.

The alloy density was 1.78 g/cm^3^. The alloy was produced in the Special Alloys Development Laboratory, Faculty of Materials Science and Engineering, Polytechnic University of Bucharest, Romania. The alloy is composed of magnesium 99% wt and calcium 1% wt. The Mg–1Ca alloy material was obtained using compact magnesium (99.7% purity) and granules of calcium (99.8% purity) in a rocker crucible furnace with sulphur hexafluoride (to protect against oxidation). For melting, a CT-AL-1.1 electric crucible furnace (Protech, Zhengzhou, China) with a graphite crucible with kantal resistors, automatic temperature control, and instant thermocouple was used. The ingots were obtained by casting. The prepared bulk Mg–1Ca alloy was cut into shaped pieces and the sample’s surface was polished and cleaned without any special surface pre-treatment.

All implants were chemically sterilized by means of ethylene oxide. An Amsco Steris 3017 EO-Sterilizer was used, with a 16 h sterilization cycle at 54 °C and negative pressure. The method was chosen because autoclave sterilization, i.e., vapors under pressure, will lead to a rapid oxidation of Mg–1Ca alloy, and standard thermal sterilization methods such as hot air oven, can induce spontaneous combustion of the Mg component of the alloy. This method of chemical sterilization with ethylene oxide, addressed to magnesium alloy, is a safe, efficient, and relatively inexpensive method.

Rabbits were all weighed preoperatively, and the values are shown in Table 1. Also all the cases have been clinically evaluated according to FELASA standards (Table 2) [17]. The evolution of weight and clinical status were recorded both before surgery (BS) and after surgery (AS) at 2, 4, and 6 weeks, respectively.

Based on this data, the “human objectives” of the study were established, representing an indication for early euthanasia of the animal to reduce its suffering. Thus, a 20% weight loss, equivalent to 20 points, was considered a severe degradation of the animal’s condition, and therefore an indication for euthanasia [18].

The day before the surgery, blood samples were taken using the auricular artery as a source. Blood tests are necessary to assess the possible hemolytic effect of Mg–1Ca alloy, possible interference with coagulation processes, changes in inflammatory, complete blood count (CBC), and serum electrolytes (SE) markers, impairment of vital organ function, kidney, liver. All blood markers determinations were performed using authorized animal laboratories, with physiological values of parameters being provided (Table 3). Follow up of blood markers was performed 6 weeks after surgery.

Before surgery, prophylactic antibiotic therapy was used with a single 10 mg/kg/bw subcutaneous dose of Enrofloxacin (Boytril 5%), to combat sepis [19]. Prophylactic antibiotic therapy was also used for two days after surgery, with the same drug, in a dose of 5 mg/Kg/bw q12 h [19]. Sepsis, in the postoperative stage, would render completely ineffective the alloy integration. Postoperative pain was controlled using Meloxicam, at a dose of 0.2 mg/Kg/bw q24h. The administration period was three days.

Surgery was performed under general inhalator anesthesia using a mask for veterinary use, with an AEON-7400A anesthesia machine (Figure 2). The anesthetic agent used was Sevoflurane. Induction was achieved by subcutaneous administration of 20 mg/Kg/bw Ketamine.

In the case of group one, the cylindric implant was inserted intramedullary in the tibia, through an anterior approach, supra tibial tuberosity (Figure 3a).

In the case of group two, the parallelepipedal implant was inserted through a thigh lateral approach, in the interstice between the vastus lateralis and the vastus intermedius muscle (Figure 3b).

The local evolution of the implants, and of the tissues adjacent to the implant, was followed radiologically. Radiographs were taken at 2, 4, and 6 weeks after surgery.

The rabbits were euthanized 6 weeks after implantation. To euthanize the rabbits without causing suffering, we chose as initial procedure to anaesthetize them by subcutaneous injection of Ketamine at a dose of 20 mg/Kg/bw. Euthanasia was performed by injecting T61 (Ethambutramide + Mebezonium Iodide + Tetracaine hydrochloride) in a dose of 1 mL intravenously.

Biopsy samples were harvested from muscle tissue and bone tissue in close proximity of the implants and within the contact area. Tissue samples were sent to pathology for histological examination. The objectives of the histological examination were to highlight the cell viability and morphology of bone or muscle cells in the immediate vicinity of the Mg–1Ca alloy implant and gas bubbles.

In the same time tissue samples were harvested from vital organs, liver, kidney, and lung for histological examination. The objective was to identify possible embolic lesions, or changes in tissue structure, hypotheses raised by in vitro studies.

Samples of muscle tissue and vital organs were fixed in standard paraffin blocks. Using a microtome, sections of 4 µm thickness were cut and stained with Hematoxylin–Eosin (HE). Bone tissue samples were decalcified in 10% EDTA solution. After decalcification they were fixed in paraffin blocks, from which 4 µm thick sections were cut and stained with Hematoxylin–Eosin (HE), Van Gieson (VG), and Masson’s Trichrome (MT) stains better highlight the cellular activity and viability of bone tissue in the vicinity of the Mg–1Ca alloy implant.

For the statistical determinations of the differences recorded by the SE parameters, we used the Mann–Whitney–Wilcoxon signed rank test for two paired samples instead of the t-test, due to the small number of subjects. The Wilcoxon test uses nonparametric procedures, using the differences between paired values and their ordering, being an ordinal data test. All details were integrated into an Excel database. Standard statistical reports have been created using the Excel included tools. For advanced statistical analysis we used IBM`s SPSS, Statistics Campus Editions version (SPSS Inc., Chicago, IL, USA) with university account (https://www.ibm.com/products/spss-statistics-campus-editions). Statistical significance value was set for *p* < 0.05.

## 3. Results

### 3.1. Radiological Followup

Radiological images were taken postoperatively, with follow-ups at 2, 4, and 6 weeks after surgery for each group.

Figure 4 shows coronal and lateral radiographic images taken the day after surgery, in group one. The Mg–1Ca alloy pin inserted intramedullary into the tibial diaphysis is marked with the number 1.

Figure 5 shows coronal and lateral radiographic images taken six weeks after surgery. The intramedullary Mg–1Ca alloy pin, partially resorbed is marked with number 1. The number 2 indicates intra-articular gas bubbles of hydrogen, indicating pneumarthrosis in minimal quantity. The number 3 marks the bone tissue adjacent to the implant, with normal to opaque appearance, without radio-transparent lines, delimiting the implant, lines that could suggest the development of fibrous tissue at the bone-implant interface. The radiologic aspect of the austenitic steel implant is used for comparison (Figure 6).

In rabbit case no. 1 an incident occurred, with a fracture of the tibial diaphysis (marked no 2) when the Mg–1Ca alloy implant (marked no 1) was inserted. The radiological images are shown in Figure 7. The final evolution was, however, satisfactory, with the development of bone callus (marked with number 3), evidenced radiologically at 6 weeks post-operatively, in Figure 8. The evolution of the implant and bone tissue was similar to the other subjects in the group.

Figure 9 shows coronal and lateral radiographic images taken the day after surgery, in group two. Mg–1Ca implant, inserted in the thigh muscle tissue, is marked with number one.

The local evolution, 6 weeks after surgery, is shown in Figure 10. The Mg–1Ca alloy, partially resorbed, is marked with the number 1. Around it, developed in the muscle tissue, hydrogen bubbles can be observed, marked with the number 2.

### 3.2. Clinical Evolution

#### 3.2.1. Weight Evolution

Rabbit’s weight evolution, in kg, assessed before surgery (BS) and at 2, 4 and 6 weeks after surgery (AS) is presented in Table 4 and Figure 11 for group 1 and in Table 4 and Figure 12 for group 2.

In group 1 the maximum weight loss was 14,63%, recorded in the case of rabbits with number 1, at 2 and 4 weeks after surgery followed by 10% recorded for subject no 6 (Table 4, Figure 10). In all cases a more pronounced weight loss was recorded in the first 2 weeks, the rate of weight loss decreasing towards 4 weeks. From this point onwards an increase of recorded values is observed.

In group 2 the maximum weight loss was 10.52%, recorded in rabbit number 10, at 2 weeks after surgery (Table 5, Figure 11). Again, in all cases a more pronounced weight loss was recorded in the first 2 weeks. From this time onwards, an increase in the recorded values is observed, towards values close to the initial ones.

#### 3.2.2. Clinical Score Evolution

The results of the rabbits clinical score are shown below, divided by groups. The cumulative clinical signs score had a maximum allowed value between 35 and 40 points.

In group one, the maximum score recorded was 15 points for rabbit number 1 and 6. Subject number one recorded a rapid increase up to 4 weeks, but after this period the values decreased. This rapid and long-lasting increase can be explained by the occurrence of iatrogenic fracture (Table 6 and Figure 13).

In the case of the other subjects, the maximum is reached at 2 weeks, followed by a steady decrease in values.

In group 2, the maximum score recorded was 12 points, in the case of the rabbit 10, at 2 weeks. All subjects recorded increased clinical score values at 2 weeks, and these steadily decreased by 6 weeks to values close to the initial ones (Table 7 and Figure 14).

The maximum values of the clinical scores, calculated for all subjects, represented a maximum of 43% of the maximum allowed values (35–40 points). No wound dehiscence was recorded in any case.

#### 3.2.3. Blood Loss Evolution

The values obtained by monitoring physiological blood parameters, before surgery and at 6 weeks after surgery, are summarized in Table 8 and Figure 15.

The assessed parameters, magnesium, calcium, sodium, and potassium recorded mean values, before, and at 6 weeks after surgery, that fall within the range of normal physiological values of the species. The differences recorded between the two evaluation times were minor, statistically insignificant, as shown in Table 9.

In the case of CBC, rabbit case no. 5 was analyzed separately due to low hemoglobin and hematocrit values recorded at enrolment. These in conjunction with the normal values of the red blood cell count, indicate an iron deficiency anemia.

Table 10 and Figure 16 show the evolution of the RBC count, registering a slight increase of 2.7% between the before surgery and the 6 weeks after surgery, mean values. Both values are within the normal values characteristic of the species. In the case of rabbit case no 5, we also record values in the normal range, with an increase of 12.8%.

Table 10 shows the evolution of the mean hemoglobin values recorded before surgery and at 6 weeks after surgery. The recorded values fall within the normal, physiological limits, characteristic of the species, registering an increase of 1.4%. In subject number 5, however, a decrease in hemoglobin of 7.8% was recorded.

The evolution of renal and hepatic functional markers together with alkaline phosphatase, a marker of the osteolysis process, are shown in Table 11.

Urea recorded a slightly increased mean before surgery value compared to the normal value, 24.09 mg/dL compared to 23.50 mg/dL, but at 6 weeks after surgery we recorded a mean value of 23.35 mg/dL, a value that falls within the normal, physiological limits of the species. The mean preoperative creatinine value was 0.73 mg/dL and 0.74 mg/dL at 6 weeks, both within the normal range (Figure 17).

The evolution of liver function indicators, ALT and AST are shown in Figure 18. Both ALT and AST recorded mean values before surgery and at 6 weeks after surgery in the normal range, being 25.8 IU to 23.2 IU for ALT and 23.3 IU to 24.8 for AST.

The evolution of coagulation process markers is presented in Table 12 and Figure 19. In the case of platelet count evolution, rabbits 4 and 5 were analyzed separately, due to the high platelet count values recorded at enrolment in the study, they showed a higher thrombocytosis, 1190 K/μL respectively 945 K/μL. In the case of the other 8 rabbits, the mean value recorded before surgery was 429 K/μL and at 6 weeks was 475 K/μL, the effective increase being 10.65%. Both mean values were within the normal range characteristic of the species. In the case of rabbit 4, we recorded an increase of 8.74% and for rabbit 5 of 9.20%, these being comparable with the increase in the batch that remained within the normal values. Prothrombin time values are within normal limits, both before surgery and 6 weeks after surgery.

The evolution of leukocyte and fibrinogen values, as main markers for monitoring the inflammatory and infectious activity, are shown in Figure 20. In both cases, the mean values recorded before surgery and at 6 weeks after surgery are within the normal values of the species, indicating the absence of infectious, inflammatory phenomena.

### 3.3. Histological Results

In group 1, after euthanasia, the proximal half of the tibia with the included implant was harvested *én bloc*. The results of the histological examinations are shown in the following images (Figure 21, Figure 22, Figure 23, Figure 24, Figure 25, Figure 26, Figure 27 and Figure 28). The histological images from the contrast samples are shown in Figure 29 and Figure 30.

To highlight the cellular activity and viability of bone tissue in the vicinity of the Mg–1Ca alloy implant, we performed Trichrome Masson staining technique on sections as shown in Figure 23, performed at magnitude 20 × 40. Bone cells are highlighted viable bone with intracytoplasmic nucleus, number 1. The section shows collagen fibers, stained blue, noted with number 2, arranged both around bone cells and in the bone mass, a sign of osteoblastic activity. Mg–1Ca alloy, marked by number 4, which is in the process of resorption, is directly adherent to the bone surface, with no interposing tissues at the bone-alloy interface, number 5. Near the implant, the bone tissue is arranged in parallel lines, marked 3, while at a distance, it shows normal, circular arrangement in osteons, demonstrating formation of new bone at the interface with the Mg–1Ca alloy.

In group 2, after euthanasia, muscle tissue was harvested from around the Mg–1Ca implant. Histological results are showed in the following images.

Figure 24 shows muscle tissue with embedded hydrogen bubbles. Hematoxylin–Eosin stain, optical magnification 10 × 10. Hydrogen bubbles, released intramuscularly, with white appearance, are marked with the number 1. Muscle fibers, normal color and viable appearance are marked with number 2. In the section, the nuclei of the muscle fibers are marked with number 3, colored blue, sign of cell viability.

Due to the suspicion of gas embolization in vital organs and the influence of the Mg–1Ca alloy resorption process on the function of vital organs, we performed histological sections through the organs whose function we followed, and by laboratory tests.

Figure 25 shows a section through lung tissue. The alveoli appear as empty spaces, marked with the number 1. No secretions inside the alveolus, or alveolar collapsing are seen. The interstitial spaces, marked with number 2, have a normal appearance, without interstitial infiltrate. A bronchiole is marked with number 3, with normal appearance and structure, and with number 4 the blood vessels are identified. No signs of gas embolism are observed.

Figure 26 shows a section through the kidney tissue. It has a normal appearance, with no pathological signs. Glomeruli of normal appearance and structure are marked with the number 1 and collecting ducts with the number 2.

Figure 27 shows a section through the liver tissue. The liver cells, marked with the number 1, have a viable appearance with nuclei visible inside. They are arranged in columns delineating the bile ducts, (number 2). The liver tissue shows no pathological aspects.

Since radiological images showed the development of pneumarthrosis, secondary to the release of hydrogen during the resorption process of the Mg–1Ca alloy, we decided to take tissue samples from the synovial membrane to observe a possible joint lesion. Figure 28 shows a section through the knee synovial membrane, in which a hydrogen bubble marked with the number 1 is seen centrally. The synovial membrane has a viable appearance, with synoviocytes present over the entire surface of the section, with nuclei inside, a sign of viability. Blood vessels are identified, number 4, and also, limited areas of inflammatory tissue, marked with number 3.

For comparison, we also performed histological examinations on samples from the control group. In these sections the bone formed around the austenitic steel implant has a viable appearance and structure, similar to the bone around the implant made of Mg–1Ca alloy with a key difference. The difference is given by the lack of osseointegration and osteoconduction capacity. At the interface with the steel implant, the bone presents a layer of cartilaginous modification, demonstrated by the identified islands of chondroblastic cells, an appearance that is absent in the case of the Mg–1Ca alloy.

One of the main properties of Mg–1Ca alloy is its gradual degradation, as many in vitro studies showed [20,21]. Our research also concluded that, based on the aspect of radiological images of the cases, there is a clear gradual degradation of the Mg–1Ca alloy implant. Bone tissue showed opaque areas of osteocondensation in the region adjacent to the implant, suggesting the formation of new bone at the bone-implant interface, a sign of cell viability. On radiological images, there are no radiolucent lines at the bone-implant interface suggesting lack of local osteolytic process or the development of fibrous tissue at the bone-implant interface. This is concurrent with other in vivo studies, such as [1,5,22].

Some in vitro research expressed concerns regarding the possibility of Mg–1Ca alloy to potentially interfere with callus formation at the level of fracture lines [15,23,24]. However, in our study, we see the formation of callus at the level of the iatrogenic fracture, as it is clearly visible on radiological images obtained for rabbit no. 1. This observation suggests that the biodegradation process of Mg–1Ca alloy does not interfere in a negative way with the biological process of new bone tissue formation.

Gas bubble formation is believed to present a negative influence on wound healing. The theory behind this process is that gas bubbles might induce tissue dissection and thus cause a delayed wound closing [22,25]. However, in our study, none of the cases displayed delayed wound healing, with *per secundam* closing, up until euthanasia.

Some drawbacks of Ma-1Ca alloys include degradation of the two components leading to increased Mg and Ca release from the implantation site [22,26,27]. However, continuous monitoring of Mg and Ca serum levels in our study did not record any pathological changes and the values were within limits of physiological scores for the species. Serum magnesium and calcium showed average values within normal limits 6 weeks after implantation characteristic of the species, with a minimal difference between the two evaluation times, statistically insignificant, (for Mg a *p* value of 0.285, and for Ca a *p* value of 0.799). More so, some other key serologic components, such as Na and K, were also within the normal ranges. At 6 weeks after implantation the mean values are within physiological limits, and the difference from the initial mean value is not statistically significant, (for Na a *p* values of 0.241 and for K a *p* value of 0.262).

Another concern towards Mg–1Ca alloy is its potential hemolytic effect, as some in vitro studies demonstrated [28,29]. This hemolytic effect can be demonstrated and monitored by a drop in red blood cell numbers and concentration of hemoglobin. These studies showed that hemolysis rate is above 5%. However, our study demonstrated that the RBC count registers a mean value at 6 weeks post-implantation, increased by only 2.7% compared to the initial mean value, in all studied cases. Also, the evolution of hemoglobin, at 6 weeks, recorded a mean value within normal limits for the species, increasing by only 1.4% when compared to the initial mean value. Although in the case number 5, anemic at study entry, we recorded a 7.8% decrease in hemoglobin, this development cannot be considered as secondary to a hemolytic effect of the corrosion process of the Mg–1Ca alloy as this case had normal RBC values recorded at 6 weeks and especially as this value increased by 12.8%.

Some in vitro studies acknowledged the possibility of Mg–1Ca alloys to potentially alter the coagulation process by Mg ions being released as a result of a corrosion process [8,24,26,27]. However, our study did not find such behavior of the implants, this being demonstrated in two ways: by the evolution of coagulation markers and of the platelet numbers, both major indicators of coagulation power. Except for the two subjects, number 4 and 5, who presented major thrombocytosis before surgery, the evolution of the platelet counts was within the normal range of the species, with an increase at 6 weeks of 10.65%. Although subjects 4 and 5 showed at 6 weeks values significantly increased compared to normal limits, 1304 K/μL and 1032 K/μL, respectively, the increase in platelet count was 8.74% and 9.20%, respectively, comparable to the increase recorded in the normal group. Basically, the evolution of the two rabbits was similar to that of the normal group. The mean values for prothrombin time, recorded both before surgery and at 6 weeks, also fall within the normal values characteristic of the species. This, together with the evolution of the platelet count, indicates the existence of a normal coagulation process.

As other studies showed, Mg–1Ca alloys are very well tolerated at a systemic level by the host organism [11,12,26,30]. This was also demonstrated by our study through monitoring the function of several key systems, such as the kidney and liver. Urea and creatinine values, at the beginning of the study, had mean values slightly increased compared to normal threshold (24.09 mg/dL to 23.50 mg/dL), but values normalized to 23.35 mg/dL at 6 weeks. Creatinine showed mean values within the normal, physiological limits, characteristic of the species, both initially and at 6 weeks. Similarly, liver function was normal throughout the study, ALT and AST transaminases averaged within normal limits at both time evaluation points.

All major studies, researching the integration of Mg–1Ca alloy concluded that it possesses very good tissue bonding capabilities [3,7,11,24,30,31]. Our study reached the same conclusion, as we managed to demonstrate a viable appearance of the bone tissue in the vicinity of Mg–1Ca implant, by identifying the presence of osteocytes and intracytoplasmic nuclei. Tissue organization is normal in osteons and mineralization lines visible near the bone–alloy interface indicate bone tissue activity. This also suggests that the bone formation process is not interrupted, demonstrating the osteoconductive nature of Mg–1Ca alloy. At the Mg–1Ca alloy–bone interface, direct adhesion of bone to the alloy surface is observed, with no interposed fibrous or cartilaginous tissue. This demonstrates the good osseointegration capacity of Mg–1Ca alloy. More so, bone tissue viability is confirmed through osteoblastic activity, as the Masson’s Trichrome stained sections demonstrate; this proves that collagen fibers around the bone cells and in the bone mass exist, which are signs of osteoblastic activity.

Finally, some concerns have been raised through the years that pneumarthrosis can be induced as a result of hydrogen production during the corrosion process [32,33,34,35,36,37,38]. As such, the synovial membrane can be furthermore damaged as a secondary effect to hydrogen accumulation [32,33,34]. In our study, the histological sections through the synovial membrane identified hydrogen pockets, but without any signs of tissue necrosis at this level.

## 4. Limitations of the Current Study

Even if, at a clinical level, all operated cases had good viability of the limbs, displaying no signs of movement impairment, mechanical stress load tests have not been carried out on whole bones harvested after euthanasia. To properly determine the full mechanical capabilities of the bones with alloy implants, stress load tests should be carried out to determine and compare them with normal bones of similar cases or with bones from contralateral limbs. However, the focus of the current study was to assess the histological integration of the alloys, therefore this objective will be completed on a future project designed to assess this limitation.

Although the histological modifications, observed on the histological investigation, were clear, we had no means to quantify and report these findings and translate them in a synthetic score. A qualitative method to assess such parameters exists, as shown by the studies of Zhihua Han et al. [39], however, this has been done on large scale tissue samples in humans and it was not feasible for our study. Also, using Widmer’s hot spot method to quantify histological structures, such as osteoblasts, osteocytes, etc., requires immune-staining techniques that have not been performed in our study [40].

## 5. Conclusions

In the parenchyma of the organs examined, i.e., lung, kidney, and liver, no pathological changes have been noticed. All identified structures have normal appearance, without pathological infiltrates or signs of gas embolism. This does not confirm the suspicion of pathological changes induced by corrosion products and gas embolism in vital organs.

Our study shows that the clinical use of Mg–1Ca bioresorbable alloys is safe as there were no local or systemic tissue toxicity that could affect the function of vital organs or the clinical status of the subjects.

Thus, all hypotheses on toxic effects raised by in vitro studies from the literature have been disproved by the data obtained by in vivo evaluation of the clinical effect of the use of these bioresorbable alloys.

## Figures and Tables

**Figure 1 diagnostics-12-01966-f001:**
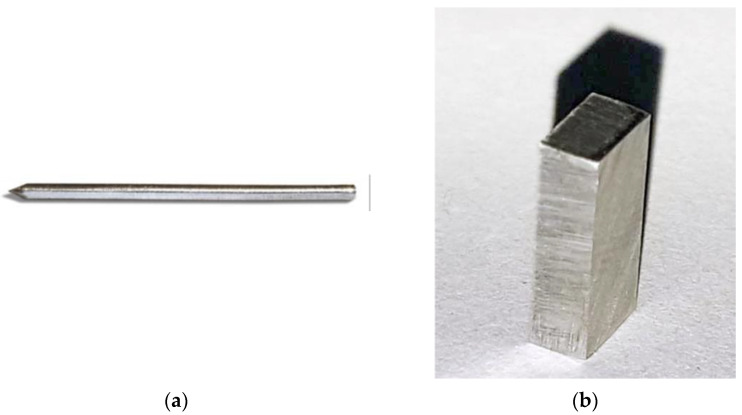
The aspect of Ma–1Ca alloy implants used in the study: (**a**) 40 × 3.2 mm cylindrical implant of Mg–1Ca alloy; (**b**) 1.2 × 1.2 × 0.2 cm parallelepiped implant.

**Figure 2 diagnostics-12-01966-f002:**
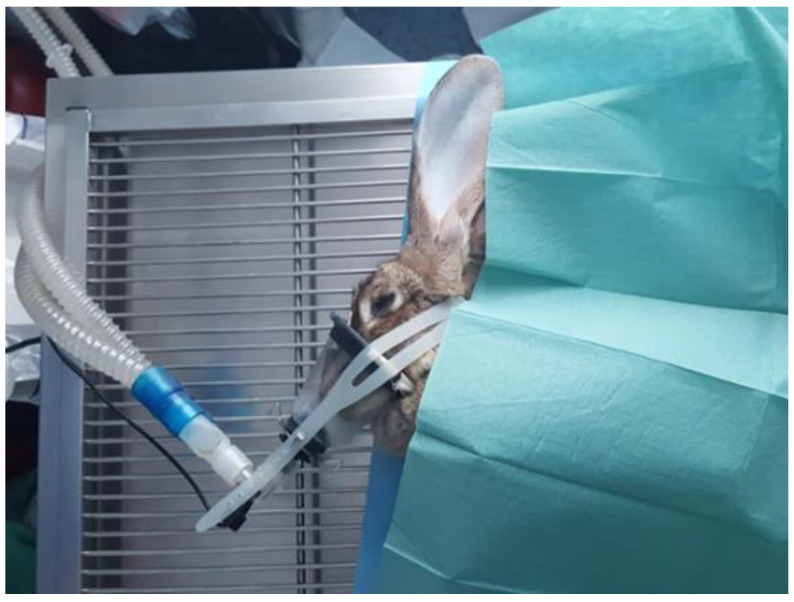
Aspect of a case under anesthesia.

**Figure 3 diagnostics-12-01966-f003:**
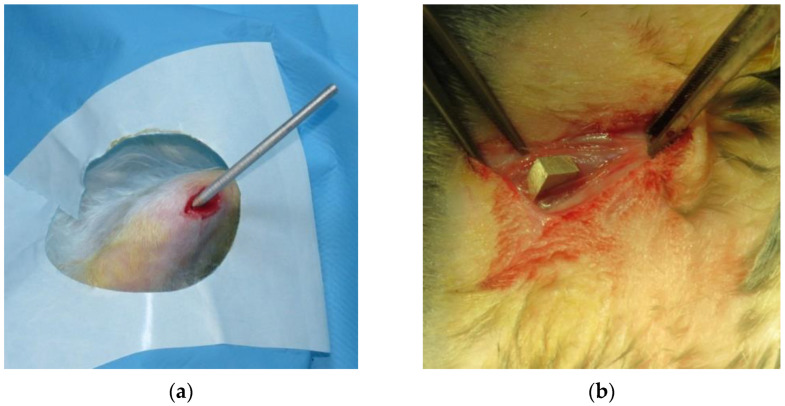
The aspects of the implants in the two groups: (**a**) group one; (**b**) group two.

**Figure 4 diagnostics-12-01966-f004:**
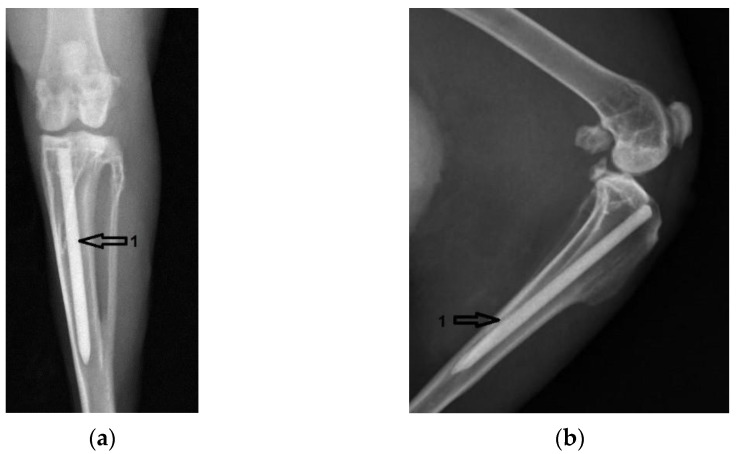
Group one, Mg–1Ca intramedullary pins, after surgery. 1-position of the Mg-1Ca pin: (**a**) Coronal view; (**b**) Lateral view.

**Figure 5 diagnostics-12-01966-f005:**
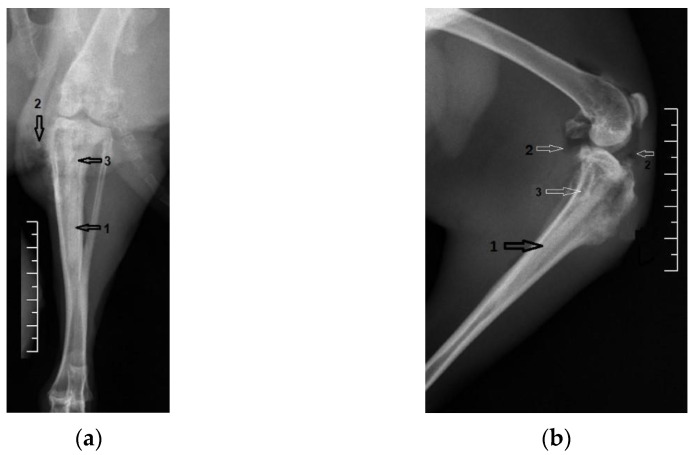
Group one, Mg–1Ca intramedullary pins, 6 weeks after surgery. 1-Mg-1Ca alloy pin with partial resorbtion, 2-Pneumoarthrosis indicated by intra-articular gas bubbles, 3-Fibrous tissue formation formed at the bone-implant interface. (**a**) Coronal view; (**b**) Lateral view.

**Figure 6 diagnostics-12-01966-f006:**
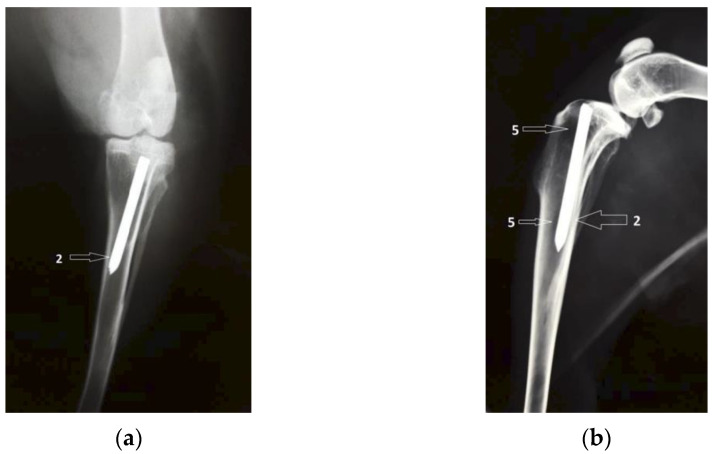
The radiological aspect of the austenitic steel rod implant, used as contrast: (**a**) the classic, inert aspect of bone-implant surface with no signs of degradation but without any sings of mineralization surrounding the implant either. 2-The austenitic steel rod implant; (**b**) on lateral incidence there are zones of increased transparency located both at metaphysis and diaphysis, indicating a possible osteolytic process, generated by the stress shielding effect. 2-The austenitic steel rod implant, 5-Zone of increased transparency, a possible osteolytic process.

**Figure 7 diagnostics-12-01966-f007:**
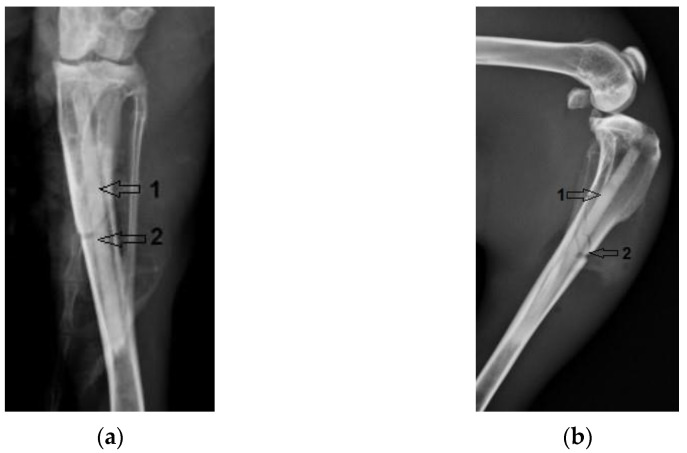
Tibial fracture, after surgery. 1-The Mg-1Ca alloy implant, 2-Fracture line in the tibial diaphysis. (**a**) Coronal view; (**b**) Lateral view.

**Figure 8 diagnostics-12-01966-f008:**
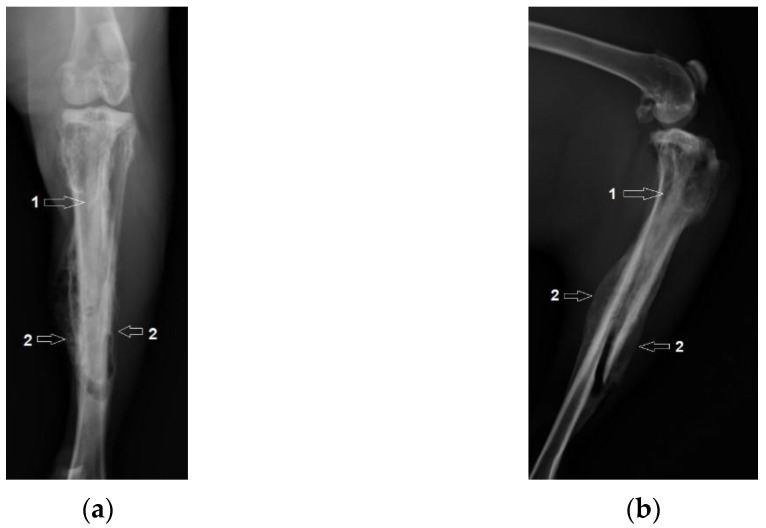
Tibial fracture, 6 weeks after surgery. 1-The Mg-1Ca alloy implant, 2-Fracture line in the tibial diaphysis. (**a**) Coronal view; (**b**) Lateral view.

**Figure 9 diagnostics-12-01966-f009:**
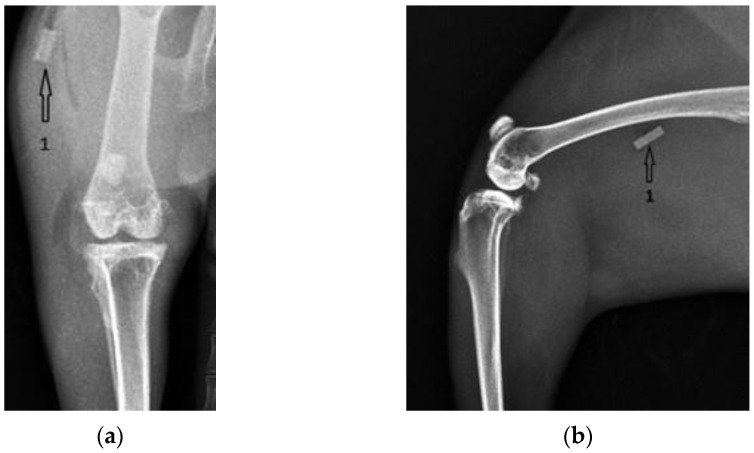
Group two, second day after surgery. 1-The Mg-1Ca implant: (**a**) Coronal view; (**b**) Lateral view.

**Figure 10 diagnostics-12-01966-f010:**
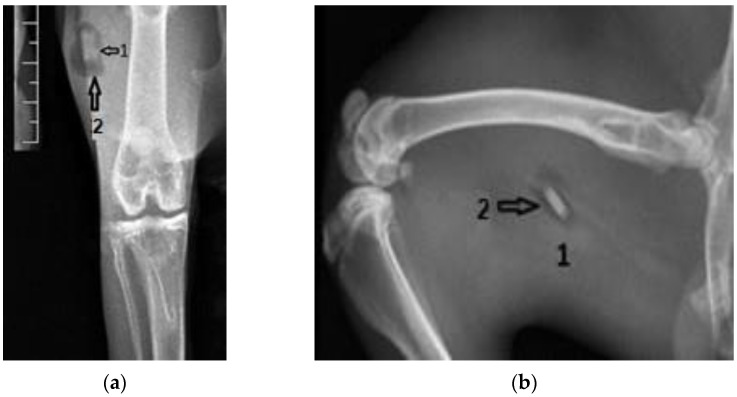
Group two, six weeks after surgery. 1- The partially resorbed Mg–1Ca alloy, 2-Hydrogen bubbles around the implant site: (**a**) Coronal view; (**b**) Lateral view.

**Figure 11 diagnostics-12-01966-f011:**
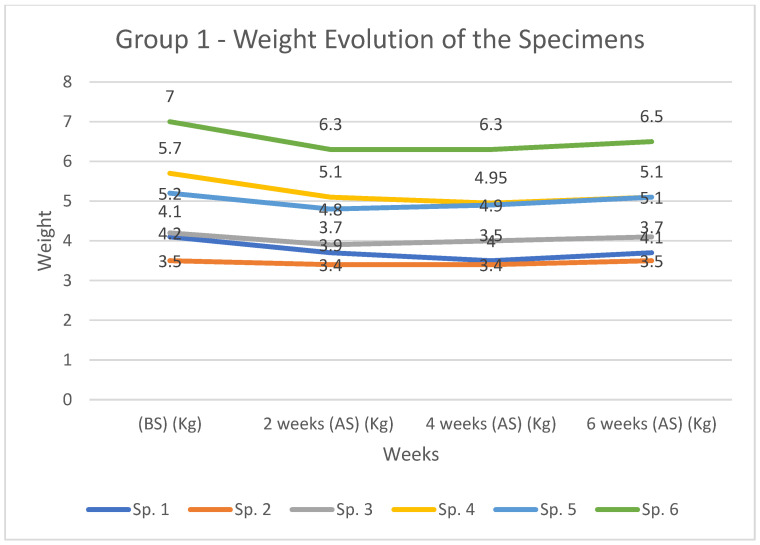
Weight distribution and comparison amongst cases from group 1 at 2, 4, and 6 weeks (AS) as well as (BS).

**Figure 12 diagnostics-12-01966-f012:**
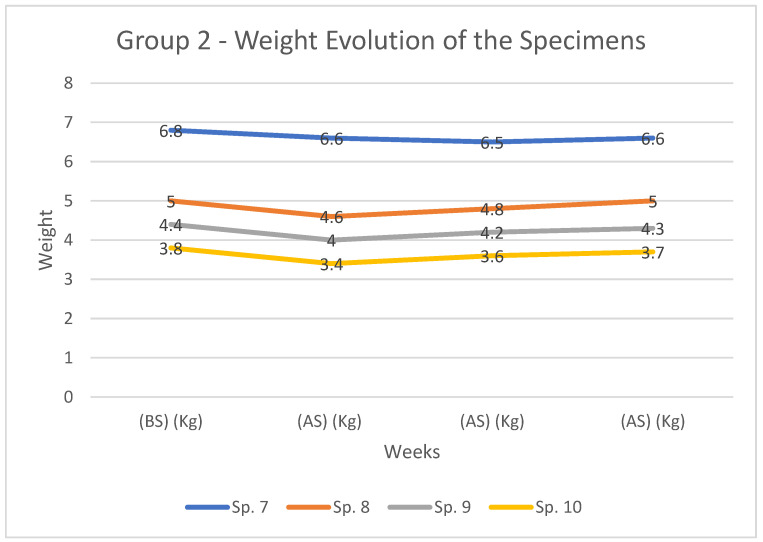
Weight distribution and comparison amongst cases from group 2 at 2, 4, and 6 weeks (AS) as well as (BS).

**Figure 13 diagnostics-12-01966-f013:**
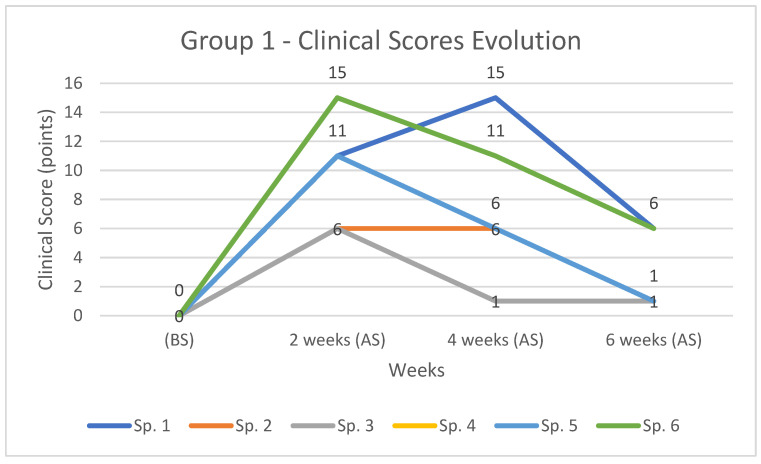
The evolution and comparison of clinical score for cases in group 1, (BS) and (AS) at 2, 4, and 6 weeks respectively.

**Figure 14 diagnostics-12-01966-f014:**
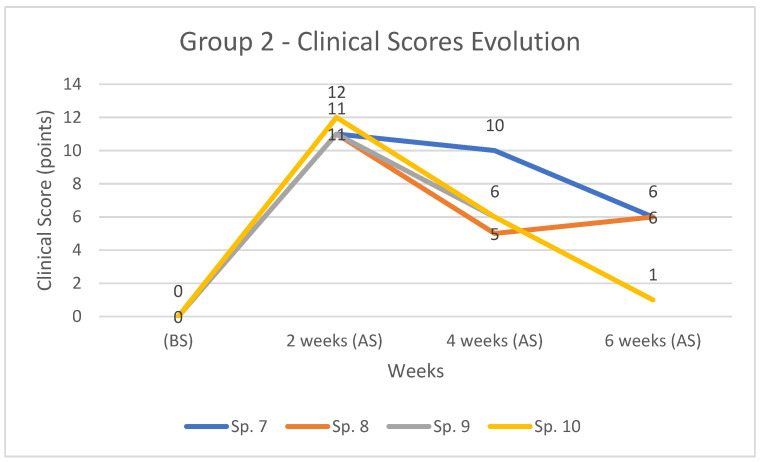
The evolution and comparison of clinical score for cases in group 2, (BS) and (AS) at 2, 4, and 6 weeks respectively.

**Figure 15 diagnostics-12-01966-f015:**
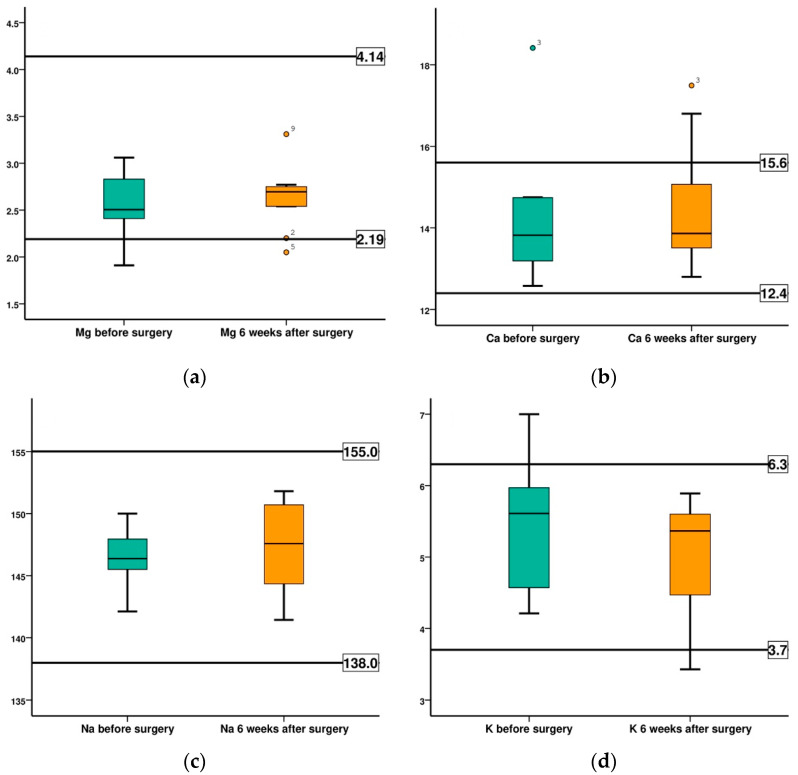
Comparison of the evolution of main serum electrolytes (SE), in both groups, before surgery (BS) and at 6 weeks after surgery (AS) for: (**a**) Magnesium (Mg); (**b**) Calcium (Ca); (**c**) Sodium (Na); (**d**) Potassium (K).

**Figure 16 diagnostics-12-01966-f016:**
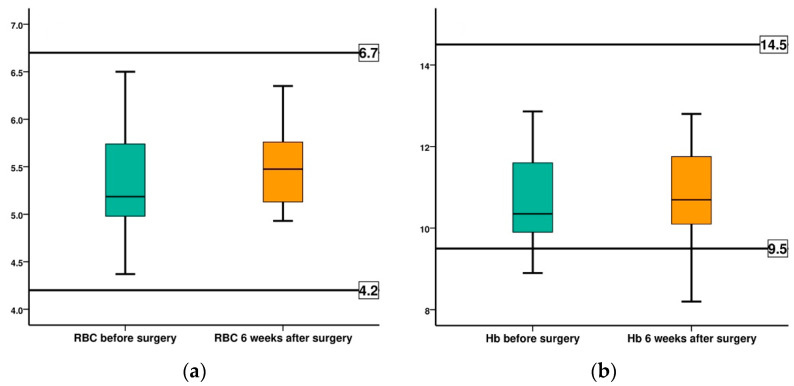
Comparison of the evolution of red blood cells (RBS) and hemoglobin (Hb), in both groups, before surgery (BS) and at 6 weeks after surgery (AS): (**a**) RBC count; (**b**) Hb.

**Figure 17 diagnostics-12-01966-f017:**
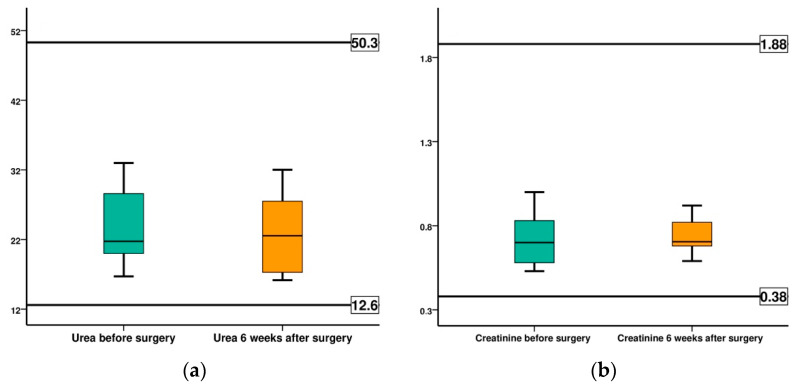
Comparison of the evolution of serum urea and creatinine, for all cases, before surgery (BS) and at 6 weeks after surgery (AS): (**a**) serum urea; (**b**) creatinine.

**Figure 18 diagnostics-12-01966-f018:**
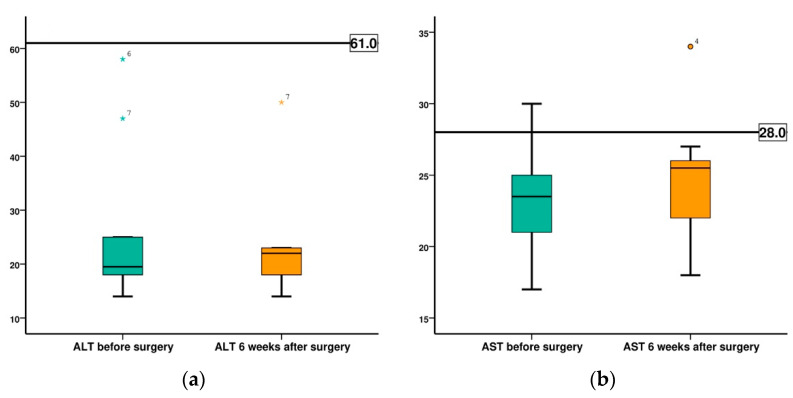
Comparison of the evolution of ALT and AST, in both groups, before surgery (BS) and at 6 weeks after surgery (AS): (**a**) ALT; (**b**) AST.

**Figure 19 diagnostics-12-01966-f019:**
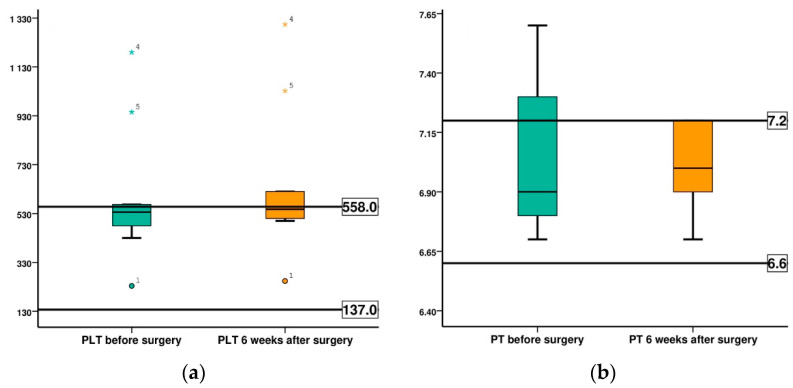
Comparison of the evolution of red platelets (PLT) and prothrombin time (PT), in both groups, before surgery (BS) and at 6 weeks after surgery (AS): (**a**) PLT; (**b**) PT.

**Figure 20 diagnostics-12-01966-f020:**
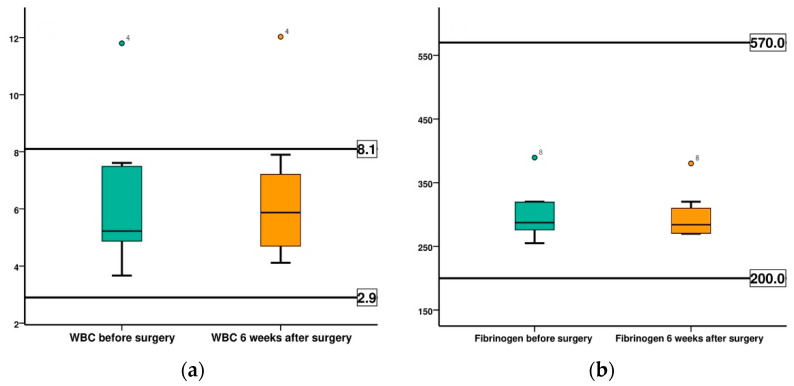
Comparison of the evolution of leukocytes and fibrinogen, in both groups, before surgery (BS) and at 6 weeks after surgery (AS): (**a**) leukocytes; (**b**) fibrinogen.

**Figure 21 diagnostics-12-01966-f021:**
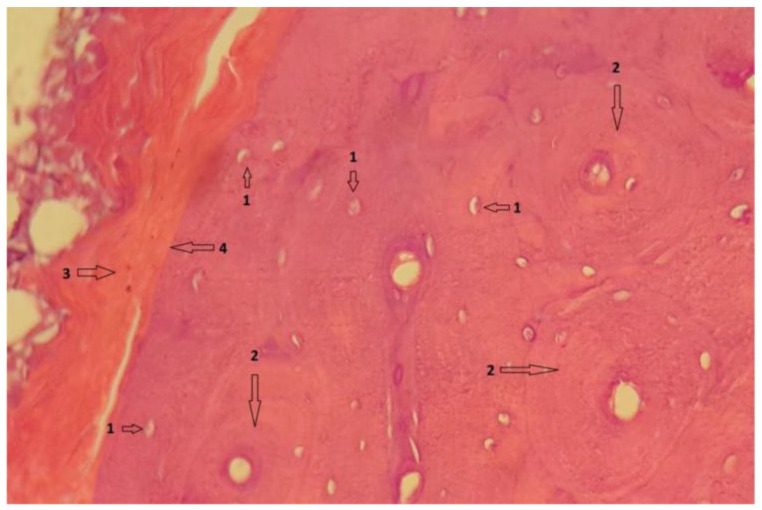
Bone section showing viable bone cells, with intracytoplasmic nucleus, visible in section, marked with 1. Bone tissue has normal configuration, being circularly arranged in osteons, structures marked with 2. Mg–1Ca alloy, degraded, is marked with 3, and the alloy-bone interface, direct without interposed tissues is marked with 4. HE stainning; 10 × 60 with digital zoom.

**Figure 22 diagnostics-12-01966-f022:**
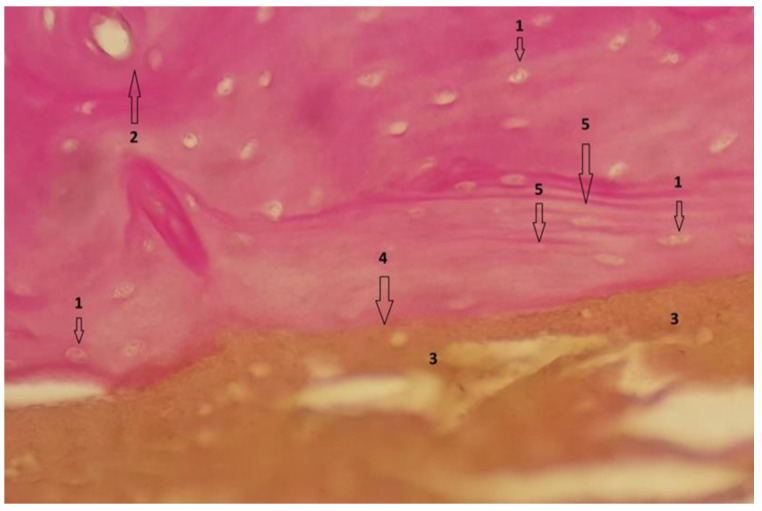
Bone tissue, showing viable bone cells with intracytoplasmic nucleus, visible in section, marked with 1. Bone tissue has normal configuration with circular arrangement of osteons, marked with 2. Mg–1Ca alloy, degraded, is marked with 3, and the alloy-bone interface, direct without interposed tissues is marked with 4. Near bone-implant interface, parallel mineralization lines are observed, no 5, sign of bone activity. Van Gieson stain; 10 × 60 magnification with digital zoom.

**Figure 23 diagnostics-12-01966-f023:**
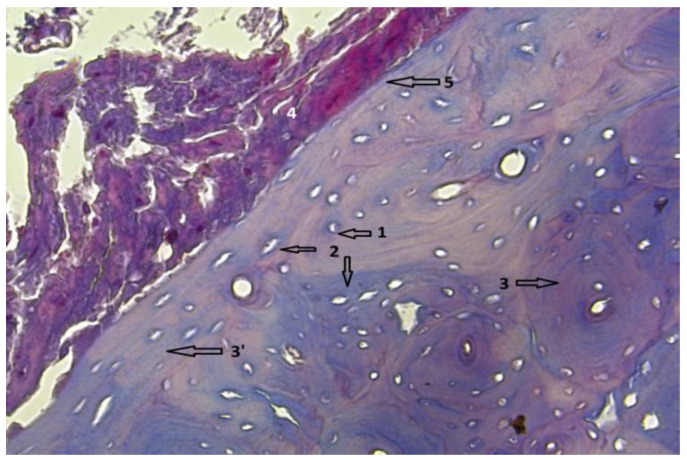
Bone tissue, Trichrome Masson stain; 20 × 40 magnitude. 1-Viable bone cells with intracytoplasmic nucleus, 2-Collagen fibers, 3-Normal bone tissue arranged in parallel line, 5-The bone-alloy interface.

**Figure 24 diagnostics-12-01966-f024:**
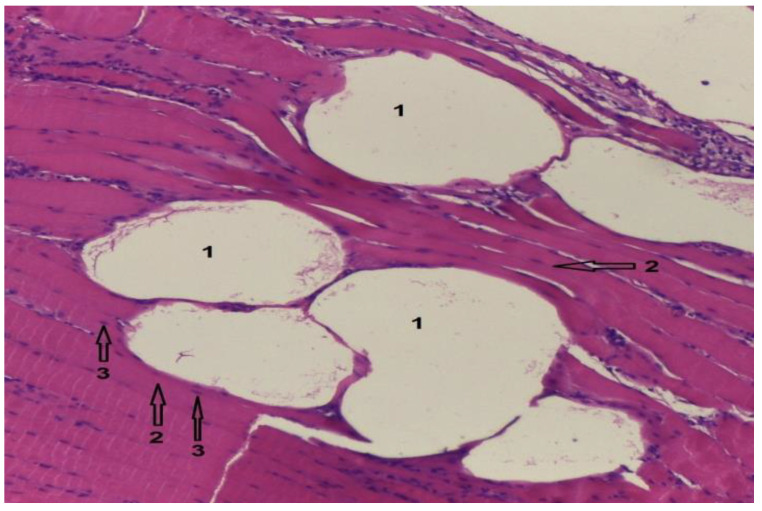
Muscle tissue, Hematoxylin–Eosin stain; 10 × 10 magnitude. 1-Hydrogen bubbles, 2-Normal muscle fibers, 3-Normal aspect of the nuclei.

**Figure 25 diagnostics-12-01966-f025:**
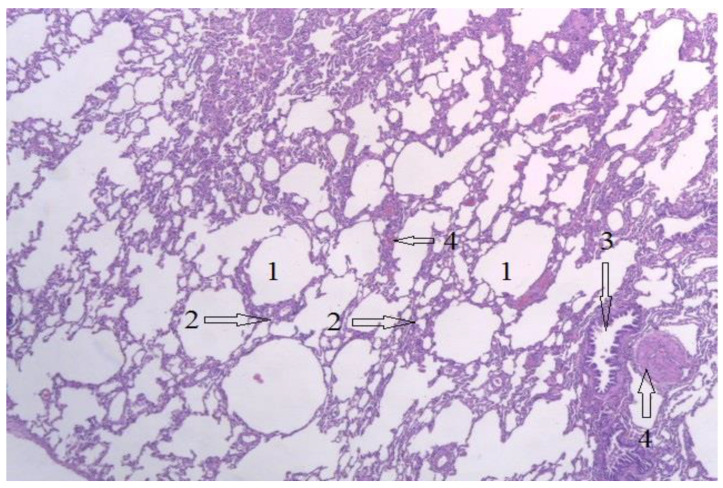
Lung tissue, H-E stain, 10 × 10. 1-Aspect of the alveoli, 2-Interstitial spaces, 3-Bronchioles, 4-Interalveolar blood vessels.

**Figure 26 diagnostics-12-01966-f026:**
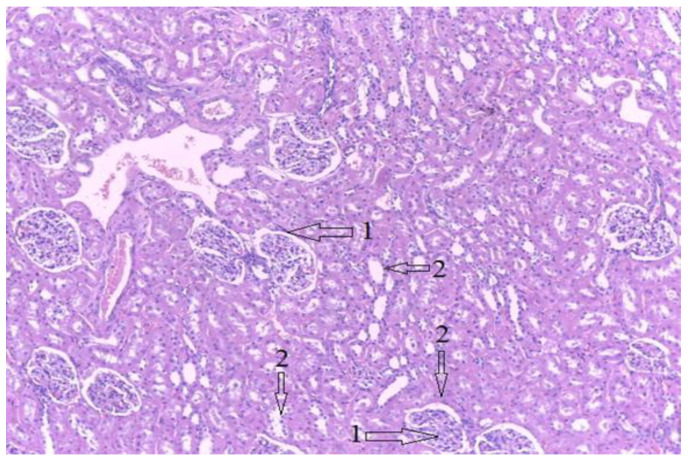
Kidney tissue; HE stain, 10 × 10. 1-Normal aspect of the glomeruli, 2-Normal collecting ducts.

**Figure 27 diagnostics-12-01966-f027:**
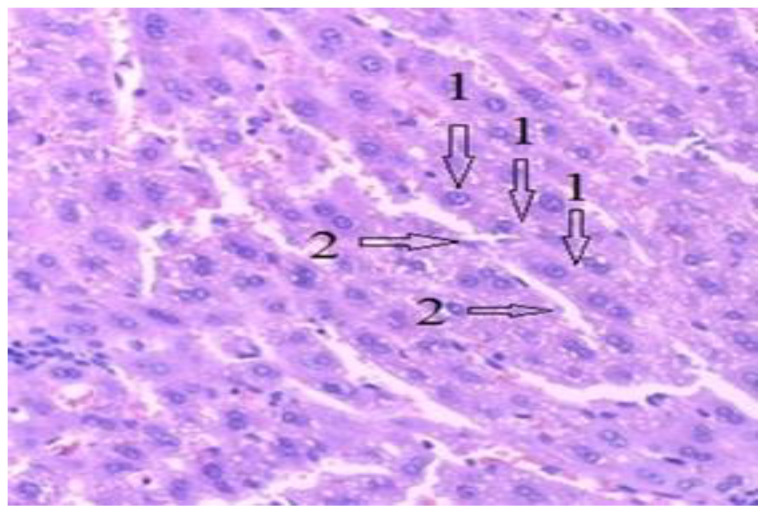
Liver tissue, H-E stain, 10 × 10. 1-Normal liver cells, 2-Normal aspect of bile ducts.

**Figure 28 diagnostics-12-01966-f028:**
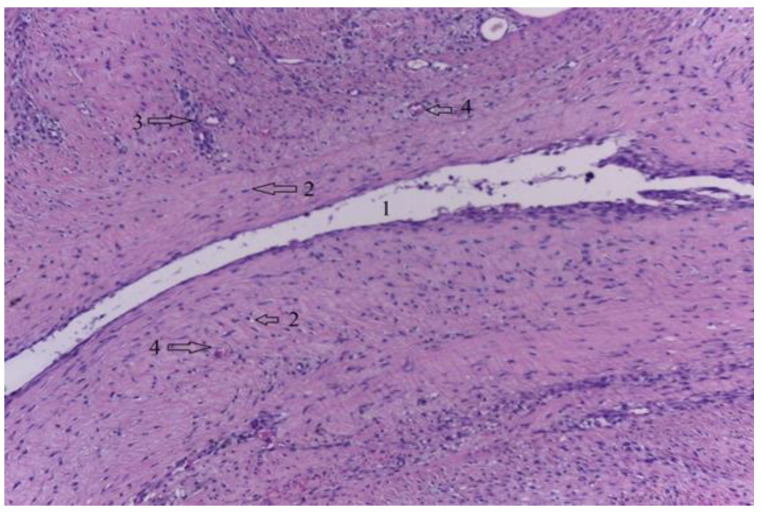
Synovial membrane, H-E stain, 10 × 10. 1-Hydrogen bubbles, 2-Normal synoviocytes, 3-Limited areas of inflammatory tissue, 4-Normal blood vessels.

**Figure 29 diagnostics-12-01966-f029:**
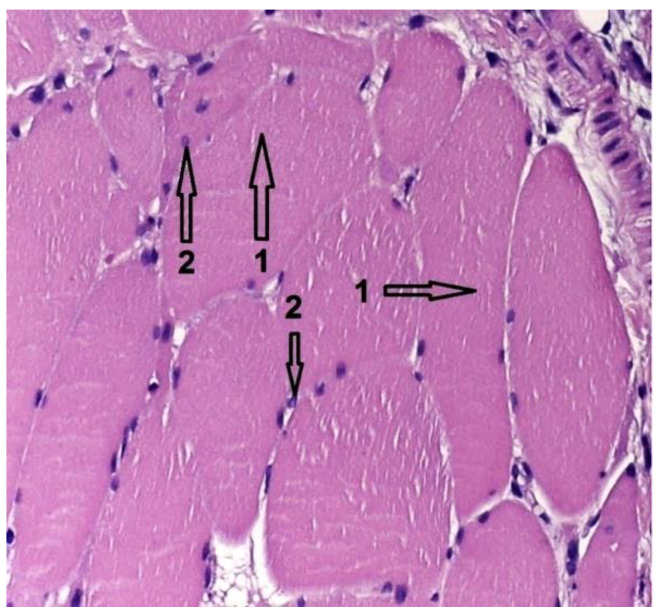
Aspect of muscle tissue harvested remotely from the implant. Hematoxilin–Eosin staining, optical magnification 10 × 20. The appearance is similar to the muscle tissue samples collected from the vicinity of the Mg–1Ca implant. 1-Viable muscle cells, 2-Normal cell nuclei.

**Figure 30 diagnostics-12-01966-f030:**
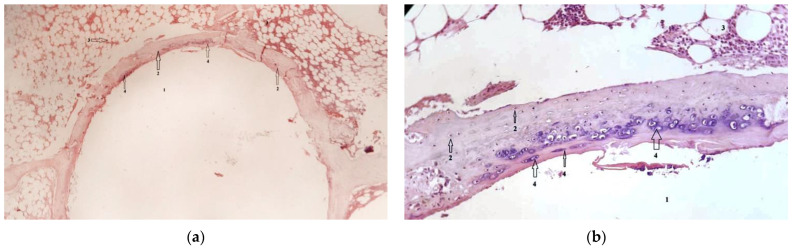
The overall aspect of the harvested bone, with the austenitic steel implant. 1-Centrally, the space left after the implant ablation. 2-A ring of bone tissue, dense, viable, with osteocytes visible in the form of discs and nuclei in their center. No remnants of alloy adherent to the bone surface are observed. At the level of the bone ring, near the implant bone interface, a chondroblastic proliferation is observed inside the bone tissue. 3-On the periphery the bone ring is surrounded by hematogenous marrow. 4-Viable basophilic cells, with a nucleus inside. These basophilic cells are identified as chondroblasts. (**a**) 10 × 4 magnification with HE staining. (**b**) 10 × 20 magnification, HE staining.

**Table 1 diagnostics-12-01966-t001:** The initial weight of the rabbits.

Specimen No.	Weight (Kg)
1	4.1
2	3.5
3	4.2
4	5.7
5	5.2
6	7
7	6.8
8	5
9	4.4
10	3.8

**Table 2 diagnostics-12-01966-t002:** FELASA clinical assement of the cases [17].

Clinical Parameter	Qualifier	Score
Weight loss	Without	0
	5%	1
	10%	5
	20%	20
Sounds made (vocalization)	Without	0
	Provoked	5
	Unprovoked	10
Posture	Normal	0
	Kyphotic	10
Mobility	Normal	0
	Low	5
	Without mobility	10
Surgical incision	No dehiscence	0
	Partial dehiscence	5
	Total dehiscence/pathological secretions	10

**Table 3 diagnostics-12-01966-t003:** Blood markers of the cases.

Class	Element	Values
CBC	Red blood cells (RBC)	4.2–6.7 (M/μL)
	Hemoglobin (HGB)	9.5–14.5 (g/dL)
	Hematocrit (HCT)	27.1–45.9 (%)
SE	Magnesium (Mg)	2.19–4.14 (mg/dL)
	Calcium (Ca)	12.4–15.6 (mg/dL)
	Seric iron	111.7–329.6 (µg/dL)
	Sodium (Na^+^)	138–155 (mmol/L)
	Potassium (K^+^)	3.7–6.3 (mmol/L)
Coagulation markers	Thrombocytes (PLT)	137–558 (K/μL)
	Prothrombin Time (PT)	6.6–7.2 sec
	Partially Activated Thromboplastin Time (aPTT)	20–22 sec
Inflammatory markers	Leukocyte (WBC)	2.9–8.1 (K/μL)
	Fibrinogen	200–570 (mg/dL)
Functional markers	Serum Urea	12.6–50.3 (mg/dL)
	Creatinine	0.38–1.88 (mg/dL)
	Alanine transaminase (ALT)	<61 (UI/L)
	Aspartate aminotransferase (AST)	<28 (UI/L)
	Alkaline phosphatase (ALP)	<397 (UI/L)

**Table 4 diagnostics-12-01966-t004:** Weight distribution of the cases, before and after surgery, in group one.

Specimen No.	Before Surgery (BS) (Kg)	2 Weeks after Surgery (AS) (Kg)	4 Weeks after Surgery (AS) (Kg)	6 Weeks after Surgery (AS) (Kg)
1	4.1	3.7	3.5	3.7
2	3.5	3.4	3.4	3.5
3	4.2	3.9	4.0	4.1
4	5.7	5.1	4.9	5.1
5	5.2	4.8	4.9	5.1
6	7.0	6.3	6.3	6.5

**Table 5 diagnostics-12-01966-t005:** Weight distribution of the cases, before and after surgery, in group two.

Specimen No.	Before Surgery (BS) (Kg)	2 Weeks after Surgery (AS) (Kg)	4 Weeks after Surgery (AS) (Kg)	6 Weeks after Surgery (AS) (Kg)
7	6.8	6.6	6.5	6.6
8	5.0	4.6	4.8	5.0
9	4.4	4.0	4.2	4.3
10	3.8	3.4	3.6	3.7

**Table 6 diagnostics-12-01966-t006:** Clinical score distribution across group one.

Specimen No.	Before Surgery (BS)	2 Weeks after Surgery (AS)	4 Weeks after Surgery (AS)	6 Weeks after Surgery (AS)
1	0	11	15	6
2	0	6	6	1
3	0	6	1	1
4	0	15	11	6
5	0	11	6	1
6	0	15	11	6

**Table 7 diagnostics-12-01966-t007:** Clinical score distribution across group two.

Specimen No.	Before Surgery (BS)	2 Weeks after Surgery (AS)	4 Weeks after Surgery (AS)	6 Weeks after Surgery (AS)
7	0	11	10	6
8	0	11	5	6
9	0	11	6	1
10	0	12	6	1

**Table 8 diagnostics-12-01966-t008:** Evolution of the SE, before and at 6 weeks after surgery.

Parameter	Time	Rabbit 1	Rabbit 2	Rabbit 3	Rabbit 4	Rabbit 5	Rabbit 6	Rabbit 7	Rabbit 8	Rabbit 9	Rabbit 10
Mg (mg/dL)	BS	2.41	2.99	2.55	2.44	1.91	2.40	2.46	2.76	3.06	2.83
	6 weeks AS	2.72	2.20	2.67	2.77	2.05	2.75	2.57	2.74	3.31	2.54
Ca (mg/dL)	BS	14.75	13.78	18.41	14.74	13.19	12.58	12.89	13.86	14.32	13.58
	6 weeks AS	13.75	16.80	17.49	15.07	14.51	12.80	13.51	12.97	13.98	13.72
Seric iron (µg/dL)	BS	185.51	109.86	111.41	130.69	93.28	123.45	115.34	127.80	130.62	129.40
	6 weeks AS	180.2	175.2	120.8	140.65	131.82	127.8	113.74	130.2	126.75	130.45
Na^+^ (mmol/L)	BS	146.31	150	147.95	145.50	148.20	145.98	143.56	146.43	147.80	142.12
	6 weeks AS	151.80	151.20	150.70	150.20	144.55	141.43	144.34	145.67	149.50	143.65
K^+^ (mmol/L)	BS	7	4.24	6.13	5.88	5.41	5.97	5.81	4.57	4.21	5.27
	6 weeks AS	3.43	4.47	5.89	5.76	5.34	5.60	5.52	4.79	4.30	5.39

**Table 9 diagnostics-12-01966-t009:** Statistical comparison of serum electrolytes (SE) such as Mg, Ca, Na, and K before and 6 weeks after surgery.

Statistic Test	Mg at 6 Weeks—Mg, BS	Ca, at 6 Weeks—Ca, BS	Na, at 6 Weeks—Na, BS	K, at 6 Weeks—K, BS
Z	−1.070 ^b^	−0.255 ^b^	−1.172 ^b^	−1.122 ^c^
Asymp. Sig. (2-tailed)	0.285	0.799	0.241	0.262

^b^ Based on negative ranks. ^c^ Based on positive ranks.

**Table 10 diagnostics-12-01966-t010:** Hemogram and leucocyte evolution comparison.

Parameter	Time	Rabbit 1	Rabbit 2	Rabbit 3	Rabbit 4	Rabbit 5	Rabbit 6	Rabbit 7	Rabbit 8	Rabbit 9	Rabbit 10
Erythrocyte (RBC) (M/μL)	BS	5.50	5.10	4.90	5.25	4.37	6.50	5.12	4.98	5.84	5.74
	6 weeks AS	5.70	5.00	5.20	5.32	4.93	6.35	5.76	5.13	6.15	5.63
Hemoglobin (HGB) (g/dL)	BS	12.86	11.50	10.40	9.75	8.90	9.90	10.30	12.40	11.60	10.30
	6 weeks AS	12.80	11.75	10.20	10.00	8.20	10.10	10.79	12.60	11.65	10.60
Hematocrit (HCT) (%)	BS	35.60	29.0	30.4	32.10	27.80	38.90	32.60	25.60	33.60	35.76
	6 weeks AS	37.30	32.60	31.70	31.90	26.90	39.90	31.70	25.40	32.10	34.90
Leucocyte (WBC) (K/μL)	BS	4.87	5.67	4.77	11.80	7.61	5.44	7.49	3.67	4.98	5.01
	6 weeks AS	4.66	6.50	6.43	12.03	7.21	5.32	7.90	4.12	4.70	5.20

**Table 11 diagnostics-12-01966-t011:** The evolution of functional markers.

Parameter	Time	Rabbit 1	Rabbit 2	Rabbit 3	Rabbit 4	Rabbit 5	Rabbit 6	Rabbit 7	Rabbit 8	Rabbit 9	Rabbit 10
Serum Urea (mg/dL)	BS	16.70	28.60	33.00	32.00	22.00	20.00	28.00	21.50	18.90	20.20
	6 weeks AS	17.20	27.50	16.17	24.00	32.00	31.30	27.10	19.80	17.30	21.10
Creatinine (mg/dL)	BS	0.56	0.69	0.53	0.83	0.71	0.79	1.00	0.67	0.98	0.58
	6 weeks AS	0.59	0.72	0.70	0.92	0.69	0.82	0.71	0.65	0.91	0.68
ALT (UI/L)	BS	21.00	18.00	17.00	19.00	14.00	58.00	47.00	19.00	25.00	20.00
	6 weeks AS	23.00	19.00	23.00	14.00	18.00	22.00	50.00	18.00	23.00	22.00
AST (UI/L)	BS	22.00	27.00	25.00	30.00	17.00	24.00	19.00	21.00	25.00	23.00
	6 weeks AS	23.00	25.00	26.00	34.00	27.00	26.00	18.00	22.00	26.00	21.00
ALP (UI/L)	BS	21.45	34.55	16.00	36.89	47.76	29.03	68.80	43.60	48.36	38.75
	6 weeks AS	22.88	32.15	15.00	34.61	45.90	30.20	70.62	44.23	47.87	37.60
	6 weeks AS	16.70	28.60	33.00	32.00	22.00	20.00	28.00	21.50	18.90	20.20

**Table 12 diagnostics-12-01966-t012:** The evolution of coagulation markers.

Parameter	Time	Rabbit 1	Rabbit 2	Rabbit 3	Rabbit 4 *	Rabbit 5 *	Rabbit 6	Rabbit 7	Rabbit 8	Rabbit 9	Rabbit 10
Thrombocyte (PLT) (K/μL)	BS	234	480	550	1190	945	430	550	522	490	567
	6 weeks AS	254	550	620	1304	1032	500	545	530	510	554
Prothrombin time (PT) (s)	BS	6.9	6.7	7.2	7.3	6.8	7.3	7.6	6.9	6.9	6.8
	6 weeks AS	7	6.8	7.1	6.9	7	7.2	7.2	7.2	6.7	6.9
aPTT (s)	BS	17.9	18.4	19.3	19.8	20.1	18.2	17.5	16.6	18.9	19.1
	6 weeks AS	18.2	18.7	19.2	19.9	19.9	18.1	17.9	17.0	18.8	19.2
Fibrinogen (mg/dL)	BS	320	276	279	295.5	301.3	278	319.6	389.4	260	255
	6 weeks AS	310	270.6	290.9	283	269.9	281	320	380.2	285	270

* These cases were analyzed separately, due to the high platelet count values recorded at enrolment in the study.

## Data Availability

All published data in this paper is available upon request from the corresponding author.

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
