# Peer review of "In Vivo Study of Local and Systemic Responses to Clinical Use of Mg–1Ca Bioresorbable Orthopedic Implants"

_diagnostics, 2022, doi:10.3390/diagnostics12081966_

Round 1

Reviewer 1 Report

This study is an in-vivo study, Mg-1Ca alloys were assessed by analyzing the weight of the animals during the study, complete blood count, serum electrolytes, liver and kidney functional markers and coagulation parameters. Radiological analysis along with tissue biopsy and pathological analysis suggests that Mg-1Ca alloys are safe for use. 

There are some shortcomings which needs to be improved, as follows:

1. Avoid using so many references in section 2. Materials and Methods. There are more references in this section than in the Discussion which reflects discrepancies in manuscript writing.

2.The 'Discussion' part needs changes. It should include comparison of your result with other similar studies. The discussion part mostly consist of elaboration of the your own result.  To improve this please add adequate latest references which can explain your result. Also comparing your result with other studies to highlight the similarities and differences is important for a good manuscript and better understanding of the reader. 

3. If there is no control group in the study, will it not affect the credibility of the result? 

4. At the end of 'Discussion' also write about the limitations of this study.

Author Response

Response to Reviewer 1 Comments

Point 1: Avoid using so many references in section 2. Materials and Methods. There are more references in this section than in the Discussion which reflects discrepancies in manuscript writing.

Response 1: The number of references in this section has been reduced to a minimum. Furthermore, Section 4. Discussion has been properly redacted, with relevant references so the ration of references between section 2 and 4 is better now.

Point 2: The 'Discussion' part needs changes. It should include comparison of your result with other similar studies. The discussion part mostly consists of elaboration of the your own result.  To improve this please add adequate latest references which can explain your result. Also comparing your result with other studies to highlight the similarities and differences is important for a good manuscript and better understanding of the reader.

Response 2: Thank you for your observation. Please find the new drafted Section 4. Discussion.

Point 3: If there is no control group in the study, will it not affect the credibility of the result?

Response 3: We do have a control group, with complete information, including histological and radiological details. It is comprised of cases from the main group that received implants of simple inert surgical steel rods in the contralateral limbs. However, we chose not to include the details regarding the control group as this would have made the paper even longer, and, giving the fact that the aim of this study was to present the clinical, radiological and histological proof that the Ma-1Ca alloys are safe for in vivo usage, we decided to scrap the description of the control group. However, your point is valid and, as such, we included a complete description of the control group in Section 2. Materials and Methods.

Point 4: At the end of 'Discussion' also write about the limitations of this study.

Response 4: Please find the newly inserted Limitation of the current study subsection of Section 4. Discussion.

Reviewer 2 Report

- the material tested in vivo are the same of those previously tested in vitro, is that correct? What kind of sterilization had been employed?

- what about power analysis to define the sufficient number of animals to employ?

- please use "case" instead of "specimens" in the text

- what about control material/group for the in vivo study? 

- the study has few number of cases and present some bias (e.g. outliers for serum markers at time zero or bone fracture during intervention). How did I you manage these with the statistical analysis of results?

- I recommend to rearrange the materials and method section to make easier the reading

- histological score about quantification of residual material/bone formation at site of implant as well as a quantitative score for cell presence should be considered 

- discussion is really poor, just repeating the results. You should discuss them critically and compare them with other relevant or similar studies present in literature.

Author Response

Response to Reviewer 2 Comments

Point 1: the material tested in vivo are the same of those previously tested in vitro, is that correct? What kind of sterilization had been employed?

Response 1: This is a grave omission from our part. We left these details out when editing the final draft of the manuscript. Please find the complete details for the sterilization method of the alloy samples received from the producer prior to implantation.

Point 2: what about power analysis to define the sufficient number of animals to employ?

Response 2: This is an observational in vivo study that tests the tissue (bone and muscular) reaction towards implants of an Ma-1Ca alloy, mimicking the environment of a human body. As this is not a large scale research that aims to establish the superiority of a method over another, we did not find appropriate to determine the power of analysis in regards to the overall number of cases.

Point 3: please use "case" instead of "specimens" in the text

Response 3: We changed all appearances of “specimens” to “cases” in the entire text.

Point 4: what about control material/group for the in vivo study?

Response 4: We do have a control group, with complete information, including histological and radiological details. It is comprised of cases from the main group that received implants of simple inert surgical steel rods in the contralateral limbs. However, we chose not to include the details regarding the control group as this would have made the paper even longer, and, giving the fact that the aim of this study was to present the clinical, radiological and histological proof that the Ma-1Ca alloys are safe for in vivo usage, we decided to scrap the description of the control group. However, your point is valid and, as such, we included a complete description of the control group in Section 2. Materials and Methods.

Point 5: the study has few number of cases and present some bias (e.g. outliers for serum markers at time zero or bone fracture during intervention). How did I you manage these with the statistical analysis of results?

Response 5: Regarding relatively low number of cases. There are a few cases of in vivo implants available worldwide, such as the one of Nina Erdmann et al (Nov. 2020) with 60 cases [Complete citation: Biomechanical testing and degradation analysis of MgCa0.8 alloy screws: a comparative in vivo study in rabbits. Nina Erdmann 1, Nina Angrisani, Janin Reifenrath, Arne Lucas, Fritz Thorey, Dirk Bormann, Andrea Meyer-Lindenberg. Acta Biomater. 2011 Mar;7(3):1421-8. doi:10.1016/j.actbio.2010.10.031]. Regarding the outliers for serum markers, could you be more specific please? They are fully commented in the Results section.

Point 6: I recommend to rearrange the materials and method section to make easier the reading

Response 6: It has been done

Point 7: histological score about quantification of residual material/bone formation at site of implant as well as a quantitative score for cell presence should be considered 

Response 7: A quantitative histological scoring system was not available in our study for 2 reasons: reason 1. a pure histological assessment system is only possible on large samples of tissues and reason 2. For minute determination methods, such as Widmer1s hot spot method, immune-staining techniques are required, that were also not possible due to the small sample size of the harvested tissue. All these have been added and explained in Limitations of the current study, just at the end of Section 4. Discussion.

Point 8: discussion is really poor, just repeating the results. You should discuss them critically and compare them with other relevant or similar studies present in literature.

Response 8: Thank you for your observation! Please find the new drafted Section 4. Discussion in the uploaded revision of the manuscript.
